# Localized Atrial Tachycardia and Dispersion Regions in Atrial Fibrillation: Evidence of Spatial Concordance

**DOI:** 10.3390/jcm10143170

**Published:** 2021-07-18

**Authors:** Edouard Gitenay, Clément Bars, Michel Bremondy, Anis Ayari, Nicolas Maillot, Florian Baptiste, Antonio Taormina, Aicha Fofana, Sabrina Siame, Jérôme Kalifa, Julien Seitz

**Affiliations:** Hôpital Saint Joseph, 26 bd de Louvain, 13008 Marseille, France; barsclement@yahoo.fr (C.B.); mbremondy@orange.fr (M.B.); anisayari84@yahoo.fr (A.A.); nicomaillot@yahoo.fr (N.M.); florian.baptiste@hotmail.fr (F.B.); antoniotaormina88@gmail.com (A.T.); aicha23@hotmail.fr (A.F.); ssiame@hopital-saint-joseph.fr (S.S.); jeromekalifa@gmail.com (J.K.); julienseitz13008@gmail.com (J.S.)

**Keywords:** atrial tachycardia, ablation, persistent atrial fibrillation, mechanism, spatiotemporal dispersion

## Abstract

Introduction: During atrial fibrillation (AF) ablation, it is generally considered that atrial tachycardia (AT) episodes are a consequence of ablation. Objective: To investigate the spatial relationship between localized AT episodes and dispersion/ablation regions during persistent AF ablation procedures. Methods: We analyzed 72 consecutive patients who presented for an index persistent AF ablation procedure guided by the presence of spatiotemporal dispersion of multipolar electrograms. We characterized spontaneous or post-ablation ATs’ mechanism and location in regard to dispersion regions and ablation lesions. Results: In 72 consecutive patients admitted for persistent AF ablation, 128 ATs occurred in 62 patients (1.9 ± 1.1/patient). Seventeen ATs were recorded before any ablation. In a total of 100 ATs with elucidated mechanism, there were 58 localized sources and 42 macro-reentries. A large number of localized ATs arose from regions exhibiting dispersion during AF (*n* = 49, 84%). Importantly, these ATs’ locations were generally remote from the closest ablation lesion (*n* = 42, 72%). Conclusions: In patients undergoing a persistent AF ablation procedure guided by the presence of spatiotemporal dispersion of multipolar electrograms, localized ATs originate within dispersion regions but remotely from the closest ablation lesion. These results suggest that ATs represent a stabilized manifestation of co-existing AF drivers rather than ablation-induced arrhythmias.

## 1. Introduction

Multiple groups have recently described patient-tailored, atrial fibrillation (AF) ablation approaches for patients in persistent AF [1,2,3,4,5,6,7]. Also named substrate-targeted approaches, these strategies differ in terms of the electrophysiological target that is selected for directing ablation, namely voltage, spatiotemporal dispersion, fractionation, or drivers. Regardless of the approach, most authors have reported the frequent occurrence of during or post-ablation atrial tachycardia (AT) episodes [3,8]. The etiology of these ATs, however, remains unclear. ATs during ablation have been viewed as a consequence of ablation [9,10] or, alternatively, as an arrhythmia relapse requiring reintervention [1,2,3,8]. Supporting the latter contention, a recent study suggested that post-ablation ATs most often originate from regions that were not previously ablated. In that study, the ECGi approach allowed for panoramic cutaneous phase mapping analysis of AF drivers and subsequent AF/AT ablation [11]. The authors suggested that post-ablation ATs exist prior to the ablation of AF drivers. When other substrate-based ablation approaches are implemented, however, the spatial relationship between AT locations and the substrate regions has been incompletely investigated. Here, we focused our attention on ATs occurring in patients undergoing ablation guided by the presence of spatiotemporal dispersion of electrograms, as described previously [3]. Specifically, we determined the location of localized ATs in regards of regions mapped as “dispersed” during AF.

## 2. Materials and Methods

We conducted a single center prospective observational study. All patients meeting the following criteria were included: (i) index dispersion-guided ablation for persistent AF between 1 November 2015 and 31 October 2016; (ii) 3D cartography performed with the CARTO 3 system.

### 2.1. Procedure

#### 2.1.1. Peri-Procedural Aspects

Oral anticoagulation was maintained before intervention. Procedures were performed under general anesthesia. After the trans-esophageal echography-guided transseptal puncture, one or several 1 mg/kg heparin boluses were administered (goal: ACT > 350 ms).

#### 2.1.2. Mapping Protocol

We used a single transseptal sheath. Mapping was performed in AF. For patients in sinus rhythm (SR), AF was induced by rapid atrial pacing using the coronary sinus (CS) catheter (from 500 to 180 ms). When AF was not inducible, isoproterenol (baseline dose: 2.4 mg/h, increased in 0.2 mg increments to reach a sinus rate > 100 beats/min) was infused. Baseline mapping in both atria was performed during AF with the PentaRay multispline catheter (Biosense Webster, Diamond Bar, CA, USA, 2-6-2 mm spacing) sequentially positioned in various regions of the RA and LA. At each location, the catheter was maintained in a stable position for a minimum of 2.5 s. As previously described [3], the operator conducted a mapping of dispersion regions. Briefly, dispersion corresponded to clusters of electrograms (EGMs), either fractionated or non-fractionated, that displayed interelectrode time and space dispersion at a minimum of three adjacent bipoles such that activation spread over all the AFCL. Dispersion regions were manually tagged on the 3D CARTO navigation system.

#### 2.1.3. Ablation Protocol

Once the initial biatrial mapping with the PentaRay catheter was obtained, there was no further analysis/evaluation performed either visually or quantitatively on electrograms recorded with the ablation catheter, nor was there any voltage analysis. Previously mapped dispersion regions were targeted with ablation, while operators did not pursue pulmonary vein isolation or lines. Radiofrequency (RF) energy was delivered (15 to 45 W) at any atrial location including the PV antrum and the CS, using an open-tip irrigated 4 mm SmartTouch SF (Biosense Webster, Inc. Diamond Bar, CA, US). No ablation was performed inside the PVs. The power was adjusted to a range of 10 to 25 W when the ablation catheter was inside the CS or on the posterior LA wall. Irrigation flow rates were adjusted according to the power delivered: 15 mL/min for >30 W and 8 mL/min for ≤30 W.

A contact force of 7 g was considered a minimum to deliver RF energy at any location. The endpoint of ablation of areas of dispersion was AF termination defined as conversion to SR or a stable atrial tachycardia (AT).

Post-ablation ATs were mapped and ablated until conversion to SR. If AF did not terminate after ablation at dispersion regions, another sequential map was obtained. When two ablated areas were in close vicinity (<1 cm) or one ablation area was adjacent to an electrically neutral structure (PV, valve), additional RF applications were performed to connect regions or regions and structures. When AF did not terminate, direct current electrical cardioversion (DCCV) was conducted (see Figure 1 for the study flow chart).

When sinus rhythm was restored by ablation, reinduction (isoproterenol and/or burst pacing) was performed at the operator’s discretion depending on the procedure duration until non-inducibility.

#### 2.1.4. AT Mapping and Classification

AT was defined as a stable organized rhythm with consistent CS activation and monomorphic P-waves. We constructed an activation map for all ATs. Entrainment maneuvers were used when needed. ATs were classified into three categories:

(i) “Localized AT” includes focal AT pattern, i.e., centrifugal activation from a discrete location, and micro-reentry pattern, i.e., >75% of the cycle length represented in the activation map at a given location;

(ii) “Macro-reentries” includes cavotricuspid dependent flutters, peri-mitral flutter, roof flutter, and other more complex macro-reentries such as double loop ATs diagnosed with activation maps and/or entrainment pacing;

(iii) “Undetermined” indicates the above maps and maneuvers were not conclusive.

The spatial relationship between localized ATs and dispersion regions is shown in
Figure 2.

Localized ATs were located according to the activation maps as well as the subsequent lesion which terminated the arrhythmia. We then assessed whether ATs were located within (AT-in, <1cm) or remotely (AT-out, ≥1 cm) from (i) a previously mapped dispersion region and (ii) the closest ablation lesion (Figure 2 and Figure 3).

## 3. Statistical Analysis

Categorical variables are expressed as *n* (%) and numerical variables as mean ± SD.

## 4. Results

### 4.1. Population

From 1 November 2015 to 31 October 2016, we included 72 consecutive patients who underwent an index procedure of dispersion-guided persistent AF ablation with CARTO system. Patient characteristics are summarized in Table 1.

### 4.2. Procedure, Mapping, and Analysis

Procedures were performed by four trained cardiac electrophysiologists in a single center: Hopital Saint Joseph, Marseille. The average procedure, fluoroscopy, and radiofrequency times were, respectively, 169 ± 37.8 ± 11 and 44 ± 25 min. The average radiation exposure was 3820c ± 4167c Gycm^2^.

Forty-seven patients (65%) were in AF at the beginning of the procedure. In the other patients, AF was induced by burst pacing and/or isoproterenol infusion.

Before ablation, the CS, RA, and LA appendage reference CLs in AF were 212 ± 59, 208 ± 58, and 201 ± 60 ms, respectively. While all patients had an LA dispersion mapping, the dispersion mapping was biatrial in 57 patients (79%). The LA and RA average volumes were 153 ± 41 and 135 ± 47 mL, respectively.

AF terminated—i.e., conversion either to AT or SR—during ablation in 64 patients (89%).

The mean procedure time to terminate AF and the mean RF time to terminate AF were 53 ± 39 and 23 ± 21 min, respectively. When AF terminated, it converted directly to SR in 15 patients (21%) and to AT in the remaining 49 patients (68%). Site of AF termination, clearly identified in 49 patients are illustrated in Appendix A. There was a subsequent SR restauration by ablation in 33 patients, and DCCV was conducted in the 16 remaining patients. Overall, there was a SR restoration by ablation in 48 patients (67%). All patients underwent an LA ablation, while 57 patients (79%) underwent a biatrial ablation. Inducibility was tested in 29 patients and achieved in 20 patients.

### 4.3. Atrial Tachycardias

We observed 128 ATs in 62 patients (1.9 ± 1.1/patient) before any ablation in patients with clinically documented persistent AF, as a transitional rhythm after ablation, or as a re-induced arrhythmia (see Figure 1—study flow-chart). Overall, 58 ATs (45%) were characterized as localized ATs (see localization in Appendix A), while 42 (33%) were characterized as macro-reentries. Of 42 macro-reentries, we observed 16 peri-tricuspid flutters (38%), 15 peri-mitral flutters (36%), and 11 LA roof-dependent flutters (26%). In 22 ATs (17%), the AT mechanism was undetermined.

### 4.4. AT before Any Ablation

Among the 72 patients referred for documented persistent AF ablation, 17 ATs were seen before any ablation, including 9 peri-tricuspid flutters, 5 localized ATs, and 3 with an undetermined mechanism.

### 4.5. Spatial Relationship between Localized ATs and Dispersion Regions

A large number of localized ATs (48/58, 83%) arose from a previously mapped spatiotemporal dispersion region. In 41/58 ATs (71%), they also located remotely from the closest ablation lesion. A representative example of how localized ATs originated within a cluster of dispersion but remotely from the closest lesion is presented in Figure 4. In this patient, the initial biatrial map highlighted the presence of multiple dispersion regions in both the RA and LA. In the RA, one small-sized dispersion region was delineated on the superior aspect of the RA posterior wall. After ablation in the LA only, AF transitioned into a stable AT, which originated from the posterior-wall RA dispersion region. This example suggests that the progressive ablation of LA dispersion regions in AF was sufficient to alter the AF dynamics into the emergence of a localized AT from a non-ablated dispersion region.

Further, we observed that, in some patients, a pattern of spatiotemporal dispersion, which is typically seen in AF, may sequentially underlie AF or AT at the same location. An example is presented in Figure 5. In this patient, prior to any ablation, AF and AT sequentially occurred while dispersion was continually recorded with a PentaRay catheter stably positioned at the posterior LA (Figure 5).

### 4.6. Redo Ablation Procedures

In total, 55 redo ablation procedures in 35 patients were performed between 16 November 2016 and 20 January 2021 (67% for AT and 33% for AF). The arrhythmia type, mechanism, and termination sites are described in Appendix A.

## 5. Discussion

Our main findings are as follows: (i) Most localized ATs (84%) recorded during dispersion-guided persistent AF ablation originated from dispersion areas mapped in AF; (ii) of these localized ATs, 72% arose remotely from the closest ablation lesion; (iii) 13% (*n* = 17) of the ATs were observed prior to any ablation.

### 5.1. Spatial Concordance between Localized AT and AF Drivers

Here, we show that there is a spatial concordance between localized AT locations and dispersion regions. Using dispersion of multipolar electrograms as a beacon to localize AF drivers, we bring evidence that ATs mostly originate from the regions where AF drivers are found. Our findings corroborate previous investigations, which found that peri-ablation ATs tend to locate where previously mapped “active” AF sites had been delineated. For example, Ban et al. [12] examined patients who experienced AT episodes after pulmonary vein isolation and CFAE-guided ablation. They demonstrated that regions exhibiting CFAEs are frequently associated with the termination of AT occurring after AF ablation. In addition, Yamashita et al. [11] investigated patients referred for persistent AF ablation consisting of the targeting AF drivers mapped with body surface multi-electrode ECG (252-lead ECGi; Cardioinsight). Among the 26 focal and 52 micro-reentrant ATs observed, 82% located in the vicinity of an AF driver. Reminiscent of some of the criteria of spatiotemporal dispersion, the sites of ATs generally presented with low-voltage, fractionated, and long-duration electrograms in AF.

### 5.2. Mechanistic Significance of Organized AT in Patients with AF

Our findings support the contention that there is a common pathophysiological mechanism that underlies AT and AF drivers. Several experimental works showed that both AT and AF may be initiated and perpetuated by wavebreaks and micro-reentrant circuits [13,14]. Other works suggested that a discrete number of co-existing drivers may perpetuate AF [15,16]. Albeit indirectly, the present observation provides additional evidence that co-existing or interchangeable AF/AT drivers originate from a unique region, which may underlie both persistent AF and ATs. In addition, our results show that some ATs might exist before AF or could alternatively represent an organized manifestation of AF. In support of these contentions, our findings suggest that spatiotemporal dispersion may represent a common electrogram footprint of AF and AT drivers (Figure 5). We also provide insight into the commonly observed phenomenon of AF transitioning into AT—and vice versa—during an ablation procedure (Figure 6).

### 5.3. Clinical Relevance in the Clinical Cardiac Electrophysiological Laboratory

ATs are highly prevalent in the population of patients with either paroxysmal [17] or persistent AF [18]. In a series of patients with AT, Israel et al. [19] concluded that most patients with a history of AF show both disorganized and highly organized AT episodes. In addition, the fact that regular ectopic beats can initiate AF episodes and play the role of a “trigger” has been abundantly demonstrated. Haissaguerre et al. [17] showed that such triggers are found in pulmonary vein regions. In addition, a focal, non-sustained monomorphic AT was seen in 40% of patients with persistent AF who underwent an electrical shock and a subsequent early AF recurrence [18]. Similarly, it is also well-known that anti-arrhythmic drugs may organize AF into a stable AT [20,21]. At the mechanistic level, Baykaner et al. [22] conducted a study in patients who underwent AF driver ablation after endocardial multi-electrode biatrial mapping using a 64-electrode catheter. The mapped AT spatially overlapped one AF source in 88% of patients, and three mechanisms were suggested: an ablation-related anchoring of AF rotor resulting in AT, a residual unablated AF source producing AT, or a spontaneous slowing of an AF rotor. In addition, Yoshida et al. [23] implemented a spectral analysis of the AF electrograms from the coronary sinus and lead V1 in patients referred for AF ablation with organization into AT during the procedure. In about half of the patients, a spectral component with a frequency that matched the frequency of subsequent AT was present in the baseline periodogram of AF. Finally, Rostock et al. [24] conducted a randomized investigation, which concluded that systematic mapping and ablation of ATs occurring during persistent AF ablation improves long-term outcomes.

Together with these studies, our work suggests that ATs could represent a so-called “simplified” manifestation of AF drivers. Thus, AF may be seen as a complex physiopathological phenomenon, whereby one or a small number of stable drivers initiate and/or perpetuate the arrhythmia. On the other hand, we acknowledge that an ablation lesion may represent an iatrogenic origin of ATs—in providing a substrate for a subsequent reentry. Our results, however, demonstrate that, during dispersion-guided ablation, RF lesions mostly cause AF driver termination, without which AT is initiated.

### 5.4. Ablation-Related ATs

Previously, ablation-related tissue injury has been shown to represent a substrate for reentry and focal discharges [25]. Karch et al. demonstrated that segmental pulmonary vein isolation leads to a higher number of post-ablation tachyarrhythmias than circumferential PVI [26]. In addition, Iwai et al. [27] showed that localized ATs during an index persistent AF ablation may differ from the ones seen in subsequent procedures and represent a poor prognosis. Such studies highlighted that ablation gaps may provide a substrate for the occurrence of localized atrial tachycardias, particularly around the PV region. In the present work, however, our observations suggest that only ~30% (17 AT episodes) of the localized ATs arose in the vicinity (within 1 cm) of the closest ablation lesion. Even if we were speculating that all these ATs were caused by prior ablative lesions, our results would still show that dispersion-guided ablation is mainly associated with non-iatrogenic localized ATs.

## 6. Study Limitations

Our work provides little information about the correlation between macro-reentrant ATs and dispersion. However, it should be mentioned that nine patients had spontaneous macro-reentries prior to any ablation. Furthermore, technical limitations prevented us from building voltage maps. In previous publications [11,22], however, localized ATs arising from AF driver sites have been associated with fragmented and low-voltage signals. Thus, future works are warranted to examine the relationship between post-ablation ATs and low-voltage regions. This study provides clinical observations and indirect evidence of a mechanistic commonality between AF and AT. Future works will need to further this investigation in examining the role of rotors and fibrotic scars in the development of AF and AT.

## 7. Conclusions

ATs are commonly observed during an index persistent AF ablation procedure guided by spatiotemporal dispersion. Our observations indicate that localized ATs mostly originate within dispersion regions but remotely from the closest ablation lesion. These results suggest a commonality of mechanism between AF and AT. Although incomplete ablation may be proarrhythmic, most ATs occurring during a dispersion-guided AF ablation procedure are unlikely to have been caused by ablation lesions.

## Figures and Tables

**Figure 1 jcm-10-03170-f001:**
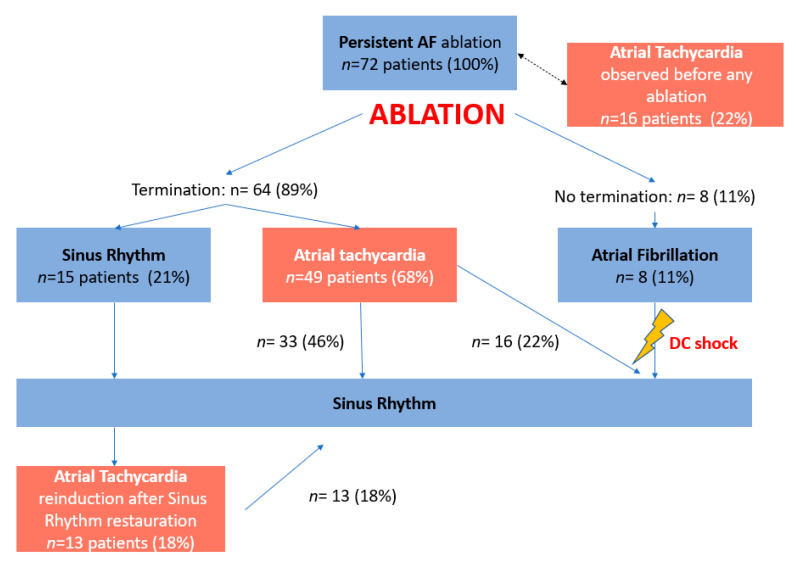
Study flow chart. AF, atrial fibrillation; AT, atrial tachycardia; SR, sinus rhythm.

**Figure 2 jcm-10-03170-f002:**
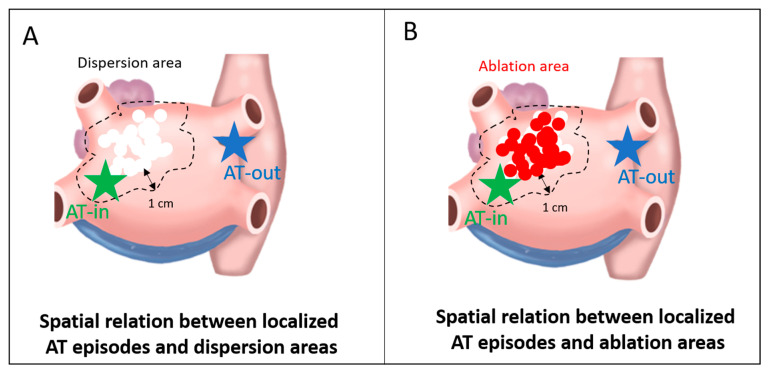
Binary classification of micro-reentrant AT localization. AT-in, localized AT located inside the boundary of the dispersion/ablation area (<1 cm); AT-out, localized AT located outside the boundary of the dispersion/ablation area (≥1 cm). (**A**) the blue star site is an example of localized AT considered distant (AT-out) from dispersion area, while the green star site is considered to be related (AT-in) to the dispersion region. (**B**) the blue star site is an example of localized AT considered distant (AT-out) from previously ablated area, while the green star localized AT site is considered to be related (AT-in) to the ablation region.

**Figure 3 jcm-10-03170-f003:**
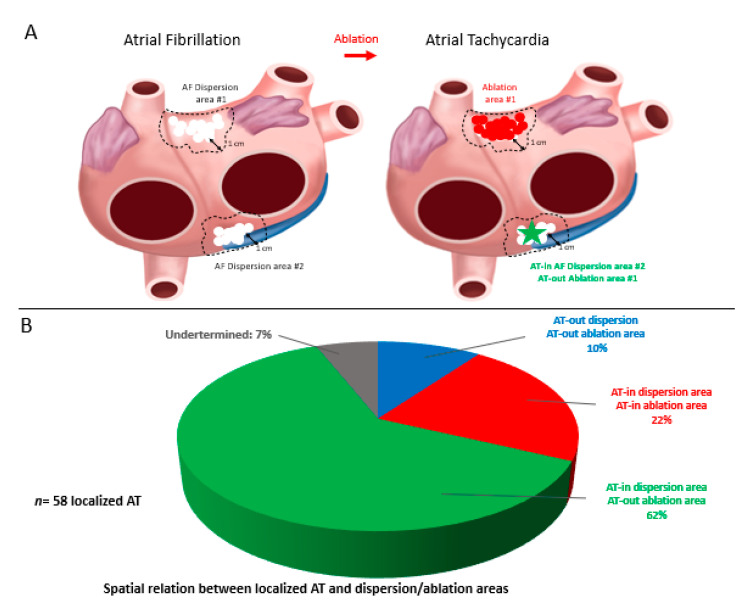
(**A**) Schematic of localized AT located in an AF dispersion unablated area. After bi-atrial dispersion mapping in AF (left picture), two regions of dispersion are highlighted. After ablation of the upper anterior dispersed site, AF turns into AT, found to be localized at the ostium of the coronary sinus (**B**). Spatial relationship between localized AT and dispersion and/or ablation areas. 62% of localized Ats were found to be in sites with dispersion pattern during AF, far from closest ablated region.

**Figure 4 jcm-10-03170-f004:**
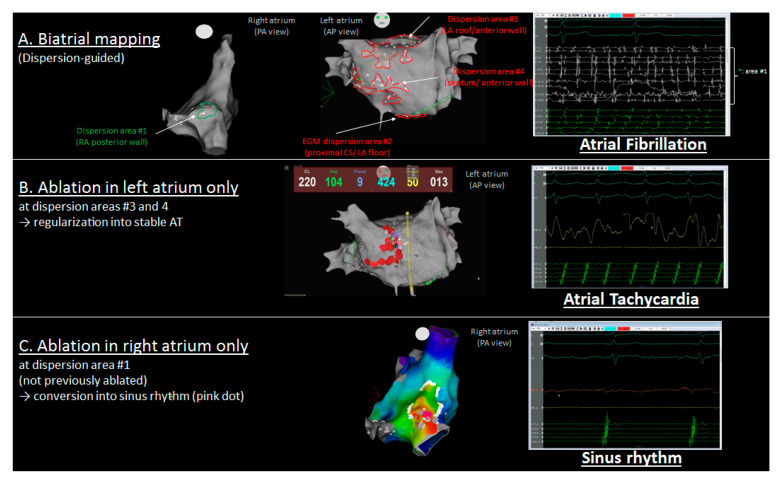
Representative example of the spatial concordance between spatiotemporal dispersion regions in persistent AF and a post-ablation localized AT emanating from the RA posterior wall, remotely from previously ablated regions. (**A**) Biatrial map showing spatiotemporal dispersion regions as encircled clusters of white dots, with the green and red lines indicating the RA and LA, respectively. (**B**) Ablation at dispersion regions in the LA led to AF regularization into a stable AT. (**C**) The localized AT presented in (**B**) terminated after ablation in the right atrium remotely from the closest ablation lesion. AF, atrial fibrillation; AT, atrial tachycardia.

**Figure 5 jcm-10-03170-f005:**
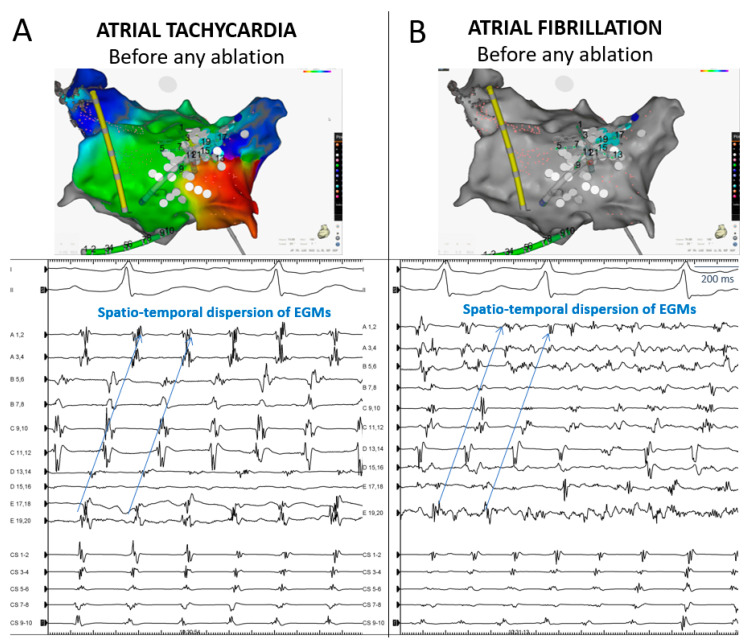
Posteroanterior 3D view of a left atrium alternatively in AT and AF prior to any ablation. The pattern of dispersion observed on a delineated posterior zone, during either AT ((**A**) the activation map on the right shows a micro-reentry at the same location) or AF ((**B**) the cluster of white tags indicates the area of dispersion beside the right pulmonary vein’s posterior antrum). AF, atrial fibrillation; AT, atrial tachycardia; AT-in, AT located inside; AT-out, AT located outside.

**Figure 6 jcm-10-03170-f006:**
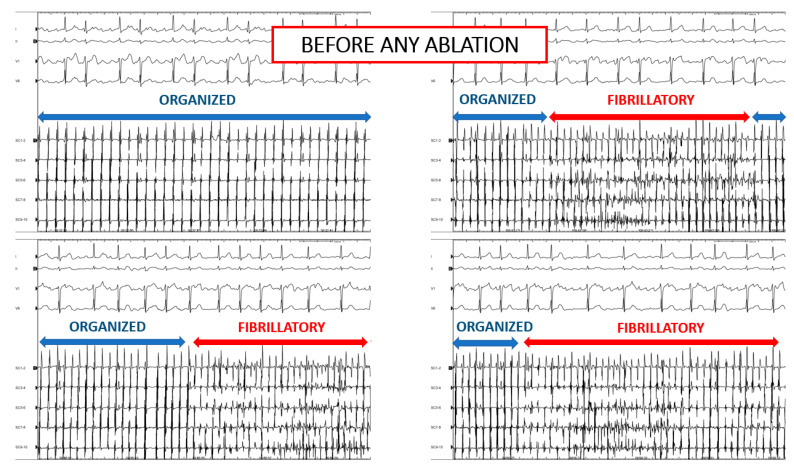
Coronary sinus EGMs showing subsequent AF and AT in a patient prior to any ablation.

**Table 1 jcm-10-03170-t001:** Study population.

Study Population	*n* = 72
Age, years	67.2 ± 9
Female	19 (26%)
AF type	
Short-standing persistent *	38 (53%)
Long-standing persistent **	14 (19%)
Persistent/unknown duration	20 (28%)
Structural heart disease	25 (39%)
Systemic hypertension	26 (39%)
Diabetes mellitus	9 (12.5%)
Obesity (BMI > 30)	4 (5.5%)
LVEF (%)	46 ± 19
LA volume, mL	153 ± 41
RA volume, mL	135 ± 47
Spontaneous AF at the beginning of procedure	47 (65%)

Values are mean ± SD or *n* (%) unless otherwise indicated. AF, atrial fibrillation; BMI, body mass index; LA, left atrial/atrium; RA, right atrial/atrium; LVEF, left ventricular ejection fraction. * When arrhythmic episodes endure from 7 days to 12 months, AF is classified as short-standing persistent. ** Continuous incidences of AF extending greater than 12 months are classified as long-standing persistent.

## Data Availability

The data presented in this study are available on request from the corresponding author.

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
