# Peer review of "Localized Atrial Tachycardia and Dispersion Regions in Atrial Fibrillation: Evidence of Spatial Concordance"

_jcm, 2021, doi:10.3390/jcm10143170_

Round 1
Reviewer 1 Report
The author should be congratulated on a well performed study of 72 consecutive patients who presented for atrial fibrillation ablation with the method of spatial and temporal dispersion. They found that, of 100 atrial tachycardia’s that were mapped and mechanism was determined, 42 or macro reentrant, And 58 were focal. Overall, 89% terminated with localized ablation and 68% of these were atrial tachycardia’s. A large portion of the atrial tachycardia‘s were found in the region of previous spatial temporal Dispersion.
dispersion.
Minor comment:
1. in the introduction, the authors state that special relationship between 80 locations and substrate regions have not been investigated. However, PM ID: 28733070 and other references assess this.
2. It would be beneficial to show cumulative regions of the ATs found
3. is there any suggestion of features of sites that are more likely to terminate or turn into an AT?
Author Response
See the attached word document.

Reviewer 2 Report
Gitenay et colleagues present their investigation study about the potential correlation between AF ablation (in Persistant cases) and recurrences of AT. Main findings: localized Ats originate from the area of ablation – mostly caused by the spotty ablation.
The manuscript is well written and has adequate length. Discussion is well elaborated.
Some concerns:
The authors cite that “Timeframe for the enrollment of these patient nov 2015 – oct 2016”
Can the authors reply about this remote timeframe? Why these patients have been remotely enrolled and immediately studied? There is no mention about the follow up. It should have been quite long (6 years) since the index procedure. It would have been nice to see at how many months these patients developed AT and in the other group how many experienced again AF.
Paragraph “Mapping Protocol”
The operator cinducted a mapping of dispersion regions as previously.
Insert the verb after “previously”
The authors state “the power was adjusted to a range of 10 to 25 W when the ablation catheter was inside 80 the CS or on the posterior LA wall”.
Could the authors specify flow rate (ml/min) of the ablation catheter during RF?
When AF terminated, it converted directly 145 to SR in 15 patients (21%)
Could the authors describe sites of termination? This would be very interesting. In terms of EGM characterization and anatomical sites. It’s not clear in your protocol if you further induced with pacing / isoproterenol infusion atrial fibrillation / atrial tachycardia.
Author Response
See the attached word document.

Round 2
Reviewer 2 Report
Accept, with the corrections.